# A Case of Food-Borne Salmonellosis in a Corn Snake (*Pantherophis guttatus*) after a Feeder Mouse Meal

**DOI:** 10.3390/ani14121722

**Published:** 2024-06-07

**Authors:** Arianna Meletiadis, Angelo Romano, Barbara Moroni, Matteo Riccardo Di Nicola, Vittoria Montemurro, Monica Pitti, Marzia Pezzolato, Elena Bozzetta, Simona Sciuto, Pier Luigi Acutis

**Affiliations:** 1Istituto Zooprofilattico Sperimentale del Piemonte, Liguria e Valle d’Aosta, Via Bologna 148, 10154 Turin, Italy; angelo.romano@izsto.it (A.R.); barbara.moroni@izsto.it (B.M.); vittoria.montemurro@izsto.it (V.M.); monica.pitti@izsto.it (M.P.); marzia.pezzolato@izsto.it (M.P.); elena.bozzetta@izsto.it (E.B.); simona.sciuto@izsto.it (S.S.); pierluigi.acutis@izsto.it (P.L.A.); 2Faculty of Veterinary Medicine, Department of Pathobiology, Pharmacology and Zoological Medicine, Wildlife Health Ghent, Ghent University, 9820 Merelbeke, Belgium; matteodinicola86@libero.it

**Keywords:** reptiles, *Salmonella*, reptile-associated salmonellosis, RAS, exotic animals, *S*. Midway, zoonosis, feeder mice

## Abstract

**Simple Summary:**

Reptiles usually carry *Salmonella* without showing any signs of infection. In this case, an adult male corn snake died 48 h after eating a feeder mouse bought online. The snake and mouse livers tested positive for *Salmonella*, specifically *Salmonella enterica subsp*. *enterica* serovar Midway. A genome analysis revealed that the two samples were from the same bacterial strain, and both had genes responsible for the bacteria’s virulence. This report is about a pet snake that acquired *Salmonella* from a feeder mouse and then died from septic shock. It shows how feeder mice can be a source of clinical salmonellosis in pet snakes, which can pose a risk to humans too.

**Abstract:**

Reptiles are usually asymptomatic carriers of *Salmonella*, with the manifestation of typical clinical signs of acute forms in adult and non-immunocompromised animals being considered exceptions. In the present case, an adult male corn snake (*Pantherophis guttatus*) was found dead due to septic shock 48 h after consuming a feeder mouse purchased online. The snake’s tissue samples and faeces were cultured for bacteria isolation. Microbiological examinations of the snake and mouse livers revealed the presence of *Salmonella enterica subsp*. *enterica* serovar Midway. A whole-genome analysis of these two isolates showed a high correlation between them: they belonged to the strain type ST-357 for the classic MLST scheme and to the strain type ST 171322 for the cgMLST scheme. Also, a virulence gene analysis revealed the presence of stdB and STM3026 genes. This report conveys a case of food-borne salmonellosis in a pet snake, transmitted from a feeder mouse, likely responsible for the snake’s death due to septic shock. It highlights the relevance of feeder mice as a source of *Salmonella* infections in snakes and the associated risks to human health.

## 1. Introduction

Reptiles are well-known *Salmonella* reservoirs [1]. They mainly act as asymptomatic carriers, harbouring multiple species of *Salmonella* in their gastrointestinal tract. These include taxa commonly associated with reptiles, such as *S. bongori*, as well as others that are not host-specific and may even be zoonotic [2,3,4]. In reptiles, *Salmonella* becomes the primary pathogen in individuals that are elderly, kept in poor conditions or experiencing a depressed immune response, and the clinical signs of salmonellosis are similar to those of many other diseases. Several authors reported that the infection in reptiles can cause hepatitis, pneumonia, septicaemia, gastritis, enteritis, osteoarthritis and sudden death [4,5,6,7,8,9,10,11]. Reports of fatal salmonellosis in various species of snakes indicated the involvement of different *Salmonella* subspecies and serovars, including *S. enterica subsp*. *arizonae*, *S. enterica subsp. diarizonae, S. enterica subsp*. *houtenae, Salmonella enterica subsp*. *enterica* ser. *Derby* and *S. enterica subsp*. *enterica* ser. Typhimurium. However, none of these reports investigated the source of infection [7,8,9,10,12,13,14,15]. 

Different reptile groups, including snakes, can also transmit salmonellosis to humans, causing reptile-associated salmonellosis (RAS) [16]. RAS, which can be acquired through direct or indirect contact with animals and their feed, can lead to diarrheal illness and serious sequelae such as meningitis and sepsis [17]. These complications can have potentially fatal outcomes, especially among infants, elderly and immunocompromised persons [18]. 

Feeder mice are among the most common feed for carnivorous snakes and in most cases are commercialized frozen rather than alive. This source has been linked to human salmonellosis, but no cases have been reported so far of fatal salmonellosis in reptiles linked to feeder mice [8,9,10,11,15,19,20,21].

This communication describes a case of food-borne salmonellosis in a corn snake (*Pantherophis guttatus*), acquired after a thawed feeder mouse meal, that likely led to death due to septic shock.

## 2. Case Report

### 2.1. Case Presentation

An adult male *Pantherophis guttatus*, approximately 10 years old, was brought by a private owner to the National Experimental Zooprophylactic Institute of Turin to investigate the cause of death. Along with the snake, a frozen feeder mouse from the same batch as the one the snake had been fed with was also submitted to the laboratory. The owner housed the snake individually in a terrarium measuring 100 × 40 × 35 cm, providing a cool area with a temperature range of 24–27 °C and a basking area of 32 °C. The substrate consisted of embossed paper, replaced almost daily. The snake’s diet typically consisted of a frozen and thawed adult mouse once every 8–10 days, with the prey thawed by soaking in hot water and then offered using clamps to simulate movement. Prior to its demise, the snake had been fed a frozen mouse from a new batch purchased online, a source the owner had not used before, and it was the sole snake in the collection fed from this batch. The owner also had three boa snakes, each housed separately in terraria, none of which showed any clinical sign. During anamnesis, the owner reported that the corn snake exhibited postprandial regurgitation and vomited the mouse 24 h after ingestion. Subsequently, over the following 24 h, the snake displayed weakness, apathy and unresponsiveness to stimuli until it was found dead.

### 2.2. Diagnostic Assessment

At the snake gross examination, abundant fat bodies were seen in the celomic cavity, and gastric mucosa and mesenteric blood vessels were hyperaemic. No other macroscopic lesions were found in the organs (Figure 1).

The small intestine was sampled for histological analyses. The tissue was immediately fixed in 10% neutral buffered formalin (4% formaldehyde), stained with haematoxylin and eosin and microscopically examined at increasing magnifications. The analysis revealed a severe diffuse necrosis of small intestine mucosa resulting in erosion of the epithelium. The necrotic mucosa was multifocally effaced by degenerate and necrotic heterophils (Figure 2). The lamina propria was oedematous and mildly infiltrated by heterophils and rare lymphocytes. Heterophils extended into the underlying tunica muscularis. Intravascular fibrin thrombi were also noted. 

Samples were taken from stomach, large and small intestine, liver, lung, heart, central nervous system, kidney and faecal samples to perform a PCR-based analysis for detection of *Ranavirus*, *Herpesvirus* spp., *Nidovirus*, *Cryptosporidium* spp. and *Chlamydia* spp. A Tissue Kit (Qiagen, Hilden, Germany) for DNA extraction was used according to the manufacturer’s instructions. Tests were performed according to protocols reported in the literature [22,23,24,25]. All PCRs resulted negative in all samples. 

Faeces were pre-enriched in Preston Broth (1:10 ratio) and then subcultured in a CCDA Agar plate for *Campylobacter* spp. isolation, both under microaerophilic and thermophilic conditions, for 48 h. No *Campylobacter* spp. was detected. 

Tissue samples taken from the liver and spleen were cultured on a Blood Agar Plate and Gassner Lactose Agar for primary culture of non-fastidious bacteria. From the liver, Enterobacteriaceae-like colonies were isolated and identified by matrix-assisted laser desorption–ionization time-of-flight (MALDI-TOF) mass spectrometry (Bruker Daltonics GmbH, Bremen, Germany) as *E*. *coli*. Given that there was no specificity of this finding, further analyses were not performed on the *E. coli*. From the spleen, no bacteria were isolated. 

Liver portions and faeces were pre-enriched in Buffered Peptone Water (1:10 ratio) and then submitted to a *Salmonella* spp. isolation standard procedure according to UNI EN ISO 6579-1:2020 [26]. *Salmonella* isolation was performed from the mouse liver using the same procedure cited for the snake samples. *Salmonella* spp. strains were isolated from the snake and mouse liver and from the snake faeces. 

The isolated *Salmonella* strains were subcultured on Columbia Blood Agar (Bec-ton & Dickinson, Dickinson, ND, USA) at 37 °C for 24 h and then serotyped according to the White–Kaufmann–Le Minor scheme, using O and H antisera (Statens Serum Institut, Copenhagen, Denmark) [27]. The *Salmonella* serotypes were identified as *S. enterica subsp*. *enterica* ser. Midway 6,14,24:d:1,7.

To investigate whether the mouse served as the source of infection for the snake, the two *Salmonella* isolates—from the snake and the mouse—were characterized and subjected to whole-genome sequencing. 

DNA was extracted starting from single colonies plated on Columbia Blood Agar (Biolife, Milan, Italy) for 16–20 h at 37 °C using the Extractme Genomic DNA isolation kit (Blirt, Gdańsk, Poland) according to the manufacturer’s protocol. DNA was quantified using a Qubit Fluorometer (Thermo Fisher Scientific, Waltham, MA, USA). After library preparation, using an Illumina DNA Library Prep Kit (Illumina, San Diego, CA, USA), genomes were sequenced using an Illumina MiSeq system (Illumina) and MiSeq V3 chemistry in a run of 2 × 151 bp paired-end reads. 

All bioinformatic analyses were performed with the tools on the Galaxy instance S.I.R.IO. http://90.147.102.165/galaxy, an online user-friendly Galaxy interface for performing bacterial genome analyses [28,29,30]. The raw reads were trimmed using Trimmomatic 0.38, by removing Nextera adaptors and other Illumina-specific sequences (Illuminaclip set to Nextera, paired-ended), by removing low-quality residues at the start and the end of the reads (leading:10 and trailing:10), by clipping reads when the average Q-scores dropped below 20 over a sliding window of four residues (slidingwindow:4:20) and by dropping reads shorter than 40 bases after processing (minlen:40) [31]. The trimmed reads were assembled de novo using Unicycler (0.4.8.0) for the bridging mode moderate contig size and misassembly rate (bridging mode set to Normal), and contigs below 200 bp in length were excluded (exclude contigs from the FASTA file which are shorter than this length, bp set to 200) [32]. Relevant assembly statistics (N50, number of contigs and median coverage against assembly) were calculated with Quast 5.0.2 [33]. The assembled genomes were processed with SeqSero 1.2 to confirm and to predict the *Salmonella* serotype, with the SPIFinder 2.0 tool to identify *Salmonella* Pathogenicity Islands (Appendix A), with MLST 2.0 for strain type identification using multilocus sequence typing (MLST) and with cgMLST Finder 1.2 for the identification of the core genome Multi Locus Sequence Typing (cgMLST) [34,35,36,37,38,39,40]. The draft genome sequences ranged from 5522 Mbp to 5530 Mbp, and the GC content ranged from 51.74% to 51.79%. The number of contigs (≥1000 bp) was 130 for both genomes, and contig N50 ranged from 107.179 to 126.847. Both the assembled genomes belonged to the strain type ST-357 for the classic MLST scheme and to the strain type ST-171322 for the cgMLST scheme. The in silico identification of the serotype confirmed the occurrence of *S. enterica subsp*. *enterica* ser. Midway.

Subsequently, an in silico identification of the presence of virulence genes in the assembled genomes of *Salmonella* was performed. To achieve this, sequences of virulence genes previously reported in the literature for *Salmonella* isolates from exotic pets and wildlife were retrieved from BLAST [41]. These sequences were then compared to the assembled genomes using the Galaxy tool “NCBI BLAST+ blastn—Search nucleotide database with nucleotide query sequence(s)” (Galaxy Version 2.14.1+galaxy2) with identity criteria set to ≥90% gene identity and a sequence length cut off of 50% [42]. Among the 196 virulence-associated genes analysed, both genomes contained the stdB and ST3026 genes of the *std* fimbrial operon.

## 3. Discussion

This report describes a case of food-borne salmonellosis in an adult corn snake associated with the ingestion of an infected feeder mouse.

The clinical evolution and laboratory results indicated that death was caused by hyperacute septic shock. The necropsy revealed the presence of gastric hyperaemia, which can be considered as a sign of acute inflammation. The absence of other lesions and the snake’s good body conditions suggest that no chronic diseases were present at the time of the examination.

Although the pathological findings were non-specific, the clinical observations and sequence of events, together with the isolation of the pathogen from both the snake and mouse, suggest that *Salmonella* was likely the causative agent of the pathology. The serotype implicated was *S. enterica subsp*. *enterica* ser. Midway, which has only been reported in reptile and wildlife sources in Canada and the United States during national surveillance programmes, but it has not yet been linked to disease in animals or humans [43,44]. 

The analysis of virulence factors in the two *Salmonella* isolates revealed the presence of the stdB and ST3026 genes of the *std* fimbrial operon. This operon is highly conserved among *S. enterica* serotypes, and it is believed to play a role in the long-term colonization of the cecum in resistant mice [45,46,47]. Additionally, the two genes are involved in the post-transcriptional regulation of the *hild* operon, which contributes to the expression of *Salmonella* Pathogenicity Island 1 (SPI-1) [48,49]. Notably, no other virulence factors, such as adhesion genes or genes responsible for immune modulation or motility, were detected. This absence may be attributed to the limited availability of sequencing data for this particular serotype, *S*. ser. Midway, which has been rarely isolated, on a global scale. Furthermore, the STM3026 gene has been identified in strains of *S. enterica* serovars Typhimurium, Enteritidis, Dublin and Heidelberg, associated with non-typhoid *Salmonella* (NTS) bacteraemia in humans in Israel [50].

The role of the feeder mouse as a possible source of *Salmonella* infection for the snake was confirmed through a whole-genome analysis of the isolates, demonstrating that they belonged to the same strain. Consequently, this case report highlights how feeder mice should be regarded as potential sources of salmonellosis, posing a direct threat not only to carnivorous reptiles fed with frozen–thawed prey but also to reptile owners through direct contact during feed preparation and handling. Indeed, reptile feeding practices should be considered as high-risk for zoonotic transmission. Numerous government institutions, including the Centers for Disease Control and Prevention, the European Food Safety Authority and the Food Standards Agency, have made concerted efforts to raise awareness among reptile owners on the risk of RAS transmission [51,52,53]. 

In humans, NTS infections often occur without symptoms and resolve spontaneously in many cases [54]. In recent years, several outbreaks of salmonellosis in humans have been reported internationally (USA, Canada and UK), with feeder mice used as reptile feed identified as the source [19,20,40,55]. 

It is important to note that in this case, the feeder mouse was provided in a frozen–thawed state rather than as live prey. Although specific details on the temperature and duration of freezing are not provided, it underscores a critical point: while frozen feed may be preferred for safety and convenience, it does not necessarily ensure pathogen-free consumption. This is particularly noteworthy when considering the widespread practice of purchasing frozen feed online of unknown source. The absence of precise information regarding the breed and freezing conditions raises concerns about potential microbial contamination in frozen feeder mice, urging reptile owners and caregivers to exercise caution and assess the reliability of their sources [56].

To the Authors’ knowledge, this represents the first evidence in Italy of feeder mice serving as a source of clinical salmonellosis in snakes. It cannot be ruled out that this batch of feeder mice may have caused other cases of salmonellosis, not only in Italy but also in other countries, given its availability for online sale. RAS poses a significant public health concern, yet its true impact remains underestimated due to the absence of specific surveillance measures in most countries [17]. Outbreaks or sporadic cases of RAS often go undetected, resulting in a gap in understanding the actual prevalence and distribution of this zoonotic infection. This case underlines once again the importance of monitoring and preventing RAS, with particular attention to feeder mice, which have the potential to trigger large, multi-state outbreaks affecting both animals and humans. 

## 4. Conclusions

This report is the first to suggest that the consumption of rodents, particularly the frozen feeder mice species, may contribute as a source of salmonellosis in snakes. Additionally, this report highlights the potential role of a specific serotype, *S*. ser. Midway, which has not previously been linked to clinical disease in either animals or humans.

## Figures and Tables

**Figure 1 animals-14-01722-f001:**
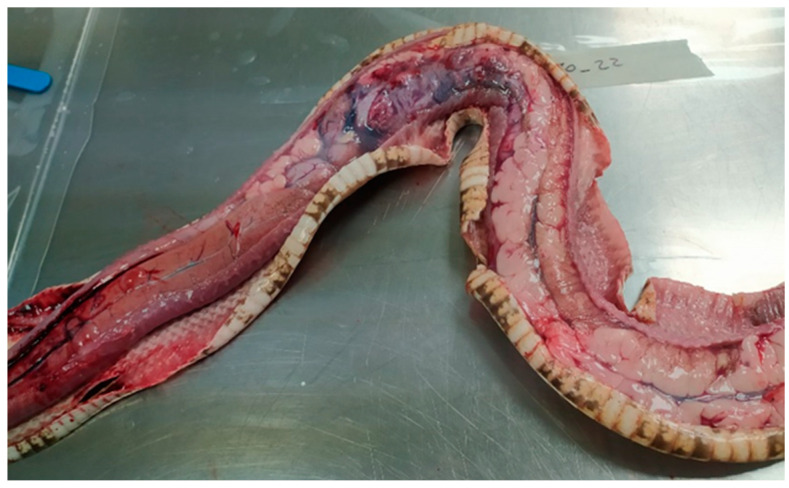
Celomic cavity at gross examination. Mild hyperaemia in the mesenteric blood vessels was observed. Fat bodies covered the coelomic cavity.

**Figure 2 animals-14-01722-f002:**
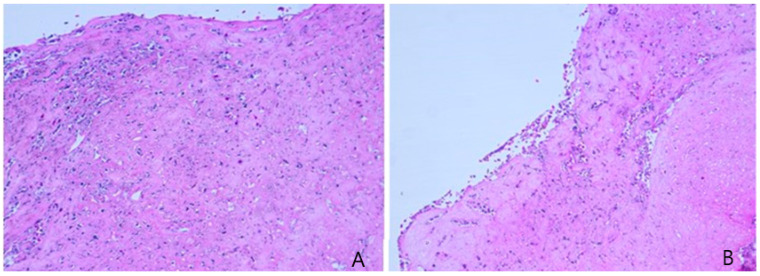
Histological section of intestinal mucosa (**A**,**B**): two different sections of necrotic intestinal mucosa with degenerated and necrotic heterophils.

## Data Availability

The datasets generated and/or analysed during the current study are available in the MendeleyData repository: https://data.mendeley.com/datasets/xcxnvbrdnp/2 (accessed on 5 June 2024).

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
