# Peer review of "A Case of Food-Borne Salmonellosis in a Corn Snake (Pantherophis guttatus) after a Feeder Mouse Meal"

_animals, 2024, doi:10.3390/ani14121722_

Round 1

Reviewer 1 Report

Comments and Suggestions for Authors

This report is about a pet snake that got Salmonella from a feeder mouse and then died from septic shock. It alerts how feeder mice can be a source of clinical salmonellosis in pet snakes which can pose a risk to humans too. So, this study highlights the danger posed by the widespread practice of reptile owners and caregivers in purchasing frozen feed online of unknown source.

This finding is not original, since several reviews and studies focusing these questions are already available (please see Gomez et al, 1997; Michael Pees et al, 2023; Ángela Galán-Relaño et al, 2023).

Specific gaps:

(i) Only an adult male corn snake (Pantherophis guttatus) was studied;

(ii) Absence of control group as reference and

(iii) In fact, methods used only detect (by clinical signs, microbiological and genome analysis), the presence of Salmonella and their virulence genes in the snake´s body as well as in the feeder mouse bought online. These findings suggest salmonellosis. It is well reported (please see Agnieszka Chlebicz and Katarzyna Åšliżewska, 2018) the potential presence of several other microbes (pathogenic bactéria) such as Campylobacter, Yersinia enterocolitica and Listeria monocytogenes in food products improper or careless processing. Efforts to detect these bacteria were not carried out.

According to the authors, the absence of precise information regarding the breed and freezing conditions raises concerns about potential microbial contamination in frozen feeder mice, urging reptile owners and caregivers to exercise caution and assess the reliability of their sources. And the danger posed by the widespread practice of reptile owners and caregivers in purchasing frozen feed online of unknown source. This study does not add great lessons, since other publications have already reinforced these points. For example, in a recente review Ángela Galán-Relaño et al (2023) proposed that at the food chain level, the prevention of salmonellosis requires a comprehensive approach at farm, manufacturing, distribution, and consumer levels. Proper handling of food, avoiding cross-contamination, and thorough cooking can reduce the risk and ensure the safety of food. Efforts to reduce transmission of Salmonella by food and other routes must be implemented using a One Health approach (Ángela Galán-Relaño et al, 2023).

Specific improvements regarding methomelody:

(a) The potential presence of several other microbes (pathogenic bacteria) such as Campylobacter, Yersinia enterocolitica and Listeria monocytogenes in food products improper or careless processing should be considered (please see Agnieszka Chlebicz and Katarzyna Śliżewska, 2018). Therefore, the confirmation of their presence in this feeder mouse batch could be checked in this study; (b) Comparison with another experimental group of snakes fed with feeder mouse batch purchased online and appropriately sterilized could also be tested.

Conclusions that are consistent with the evidence and arguments presented. The absence of other lesions and the snake's good body conditions suggest that no chronic diseases were present at the time of the examination. Observation of sequence of clinical events, together with the isolation of the pathogen from both snake and mouse, suggest that Salmonella was likely the causative agent of the pathology.

Conclusions that are not consistent with the evidence and arguments presented. Authors recognize (in DISCUSSION) that the pathological findings were non-specific and that the clinical observations and sequence of events, together with the isolation of the pathogen from both snake and mouse, suggest that Salmonella was likely the causative agent of the pathology. This sentence and this word (suggest) are not consistent with the title of this paper “A case of food-borne salmonellosis in a corn snake (Panthero-2 phis guttatus), dead for septic shock after a feeder mouse meal

In figures:

Please check in the legends of Figures

Figure 1 is Figure 2 and Figure 2 is figure 1

This study fits the scope of this journal, since, according to Instructions for authors “preference will be given to those articles that provide an understanding of animals within a larger context (i.e., the animals' interactions with the outside world, including humans”.

(1)  References should be described as follows, depending on the type of work:

  • Journal Articles:
    1. Author 1, A.B.; Author 2, C.D. Title of the article. Abbreviated Journal Name YearVolume, page range.

Comment: Please check Abbreviated Journal Names of references 2, 6, 7, 10, 12, 13, 17, 23 and 54

 (2)  References 10 and 11 are duplicated.... Please correct

(3) In INTRODUCTION: Reptiles are well-known Salmonella reservoirs [1]. [1] Geue, L.; Löschner, U. Salmonella Enterica in Reptiles of German and Austrian Origin. Vet Microbiol 2002, 84, 79–91. This reference do not correspond to this sentence. The following reference seems to be more appropriate: Michael Pees et al. Salmonella in reptiles: a review of occurrence, interactions, shedding and risk factors for human infections. Front Cell Dev Biol. 2023 :11:1251036

REFERENCES

Agnieszka Chlebicz, Katarzyna Śliżewska. Campylobacteriosis, Salmonellosis, Yersiniosis, and Listeriosis as Zoonotic Foodborne Diseases: A Review. Int J Environ Res Public Health 2018; 15(5):863)

Ángela Galán-Relaño et al. Salmonella and Salmonellosis: An Update on Public Health Implications and Control Strategies. Animals (Basel) 2023; 13(23):3666;

Gomez et al. Foodborne salmonellosis. World Health Stat Q 1997; 50(1-2):81-9.

Michael Pees et al. Salmonella in reptiles: a review of occurrence, interactions, shedding and risk factors for human infections. Front Cell Dev Biol 2023: 11:1251036;

Author Response

We thank the reviewer for his/her valuable comments. We upload the answers as a file .docx

Reviewer 2 Report

Comments and Suggestions for Authors

Many authors have proven that reptiles are asymptomatic carriers of this bacterium, but reports of the occurrence of salmonellosis in reptiles are rare. The manuscript is one such report. Based on genetic analyses, the authors showed that the source of the snake infection was mice. However, can you be sure that the cause of the snake's death was Salmonella?

The pathogen was isolated from the feces and liver of a dead snake. Why weren't other organs analyzed for Salmonella? Septic shock, indicated in the title of the manuscript as the cause of death of the snake, is diagnosed based on the presence of the pathogen in the blood. The presence of Salmonella in the feces and liver also occurs in reptiles that do not show symptoms of disease.

How much time passed from the death of the snake to the commencement of microbiological analyses?

From what biological samples were Enterobacteriaceae strains, including E. coli, isolated? Why were E. coli strains not analyzed for the presence of virulence factors. The pathogenicity of E. coli depends on the production of virulence factors, not on the production of ESBLs.

In Salmonella strains, virulence genes are located on pathogenicity islands. I suggest that the authors detect pathogenicity islands and compare the obtained isolates with the Salmonella enterica Midway genomes available in databases, e.g. GenBank: PDT000294524.2, PDT000667990.1, PDT000668846.1. I wonder whether the same virulence genes are also present in other genomes of this serovar and whether the strains represent the same sequence type (ST). I also suggest depositing the assembled Salmonella genomes in a database such as GenBank or ENA so that other authors can easily use them in their analyses.

The Discussion should include information on the asymptomatic occurrence of serovar Midway in corn snakes (cite: Pet Reptiles in Poland as a Potential Source of Transmission of Salmonella. Pathogens 2022, 11, 1125. https://doi.org/10.3390/pathogens11101125).

L81 - remove dot after "found"

L104 - remove space

L143 - in silico - italics

L216 - I suggest removing the word pathogenesis (virulence factors are responsible for pathogenesis)

Comments on the Quality of English Language

.

Author Response

(The authors gave the same response as above.)

Reviewer 3 Report

Comments and Suggestions for Authors

Although the paper is written very well, some improvements can be made:

  1. Do not use abbreviations in the simple summary, e.g., Salmonella, specifically S. Midway. It may be clearer for readers to use the full name initially and then continue with the abbreviation.

  2. The abstract effectively summarizes the key findings and significance of the study. However, it would benefit from a brief mention of the methodology used for confirming the source of infection. Please add a sentence or two on the methods used for microbiological and genome analyses in the abstract.

  3. The introduction provides a thorough background on Salmonella infections in reptiles and the risks posed by feeder mice. The literature review is comprehensive and sets the stage for the case report. However, ensure that Latin names are italicized and consistently formatted throughout all references. For example, "Salmonella enterica subsp. enterica ser. Midway" should be consistently italicized. The same issue appears in line 46 with Derby and Typhimurium.

  4. Carefully check all references in terms of species names, which should be in italics. For example, in “Geue, L.; Löschner, U. Salmonella enterica in reptiles of German and Austrian origin…,” Salmonella enterica is not in italics. Also, remove spaces between initials in papers for consistency, e.g., “Pedersen, K.; Lassen-Nielsen, A.M.; Nordentoft, S.; Hammer, A.S. Serovars of Salmonella from Captive Reptiles. Zoonoses Public Health 2009, 56, 238–242.”

With some minor revisions to improve clarity and readability, this manuscript would be a nice contribution to the field.

Author Response

Thank you for your valuable comments. All comments have been accepted and changes have been made to the manuscript in accordance with your suggestions.